# Hydrophilic Surface Functionalization of Electrospun Nanofibrous Scaffolds in Tissue Engineering

**DOI:** 10.3390/polym12112636

**Published:** 2020-11-10

**Authors:** Beata Niemczyk-Soczynska, Arkadiusz Gradys, Pawel Sajkiewicz

**Affiliations:** Institute of Fundamental Technological Research, Lab. Polymers & Biomaterials, Polish Academy of Sciences Pawinskiego 5b St., 02-106 Warsaw, Poland; argrad@ippt.pan.pl (A.G.); psajk@ippt.pan.pl (P.S.)

**Keywords:** surface functionalization, electrospinning, polymers, nanofiber, immobilization, tissue engineering

## Abstract

Electrospun polymer nanofibers have received much attention in tissue engineering due to their valuable properties such as biocompatibility, biodegradation ability, appropriate mechanical properties, and, most importantly, fibrous structure, which resembles the morphology of extracellular matrix (ECM) proteins. However, they are usually hydrophobic and suffer from a lack of bioactive molecules, which provide good cell adhesion to the scaffold surface. Post-electrospinning surface functionalization allows overcoming these limitations through polar groups covalent incorporation to the fibers surface, with subsequent functionalization with biologically active molecules or direct deposition of the biomolecule solution. Hydrophilic surface functionalization methods are classified into chemical approaches, including wet chemical functionalization and covalent grafting, a physiochemical approach with the use of a plasma treatment, and a physical approach that might be divided into physical adsorption and layer-by-layer assembly. This review discusses the state-of-the-art of hydrophilic surface functionalization strategies of electrospun nanofibers for tissue engineering applications. We highlighted the major advantages and drawbacks of each method, at the same time, pointing out future perspectives and solutions in the hydrophilic functionalization strategies.

## 1. Introduction

Tissue engineering is an interdisciplinary field of science, in which polymeric scaffolds are crucial from the biomedical perspective [1]. In this field, various forms of scaffolds might be distinguished, among which the most important are: non-injectable and injectable hydrogels, sponges, 3D printed beams, and submicron- and nanofibers [2,3,4,5]. In recent decades, electrospun nanofibrous scaffolds have gained and enjoyed the great interest of tissue engineering applications [5] with the greatest importance of nanofibers composed of aliphatic polyesters, such as poly-L-lactide (PLLA), polycaprolactone (PCL), poly (lactic-co-glycolic acid) (PLGA), and poly (lactide-co-caprolactone) (PLCL) [6]. All of these polymers are biodegradable, biocompatible, easily processable, and have appropriate mechanical properties. It is easy to control their physical and mechanical properties by tuning the polymeric solution’s concentration, using appropriate solvents and parameters of the process [7]. One of the most favorable properties of electrospun nanofibers is their morphology. It highly resembles collagen fibers in native ECM, making them attractive as scaffolds in tissue engineering, drug delivery systems, or wound dressings [8,9]. Besides aliphatic polyesters, semicrystalline fluoropolymers, especially polyvinylidene fluoride (PVDF), gained attention as smart piezoelectric scaffolds. While stress is applied to the PVDF scaffold, it generates and conducts electric signals, which induce cell regeneration [10]. This feature might be useful, especially in nervous system tissue engineering.

An appropriate scaffold formation should not only assume material morphological resemblance to the natural ECM, provide physical support for cells but mostly should contribute to cell–surface interactions [11]. Most of the electrospun polymers are rather hydrophobic, which is unfavorable from the point of view of tissue engineering. For instance, aliphatic polyesters such as PLLA or PCL show contact angles in the range of 116–135°, while for tissue engineering requirements, it ought to be below 100° [12,13]. Webb et al. [14] studied the highest level of cell attachment (NIH 3T3 fibroblasts) at hydrophilic surfaces, and the best results were observed for surfaces with contact angles in the range of 20–60°. In this regard, an appropriate hydrophilic functionalization that improves these features is thus necessary. A potential scaffold for medical applications also should have appropriate bulk properties and biological activity at the surface, but material rarely possesses both of these features simultaneously [15]. Synthetic materials have excellent bulk properties; however, their polymeric chains do not contain biologically active molecular motives. Integrin receptors bind cells with ECM, specifically, recognize Arg-Gly-Asp amino acid (RGD) sequences providing further cell adhesion to the surface. There are many natural materials, for instance, gelatin, fibronectin, laminin, or collagen, that might provide the ECM components with biological active sequences like RGD required for cell adhesion, accelerating cell growth, and providing a favorable environment for their proper functioning [13,16,17]. For this reason, it is clear that the presence of biomolecules plays a key role in promoting cell/biomaterial interactions [18].

Hydrophilic surface functionalization, followed by biomolecules attachment, can be realized by various strategies, which can be divided into three main groups: pre-electrospinning methods via bulk blending with another natural polymer [19,20], functionalization during electrospinning (e.g., coaxial electrospinning) or post-electrospinning methods such as wet chemical methods (aminolysis [6] or hydrolysis [21]), covalent grafting [22], plasma treatment [23], physical adsorption [24], or layer by layer (LBL) assembly [25]. Contrary to pre-electrospinning methods, most of the post-electrospinning methods alter fiber surface only, which is preferred from the perspective of scaffolds final properties [19].

This comprehensive review aims at an in-depth discussion on post-electrospinning hydrophilic surface functionalization strategies of polymeric nanofibers with an emphasis on their advantages and drawbacks, and the potential in further processing for tissue engineering applications. In this article, we present the electrospun nanofibers processing from both the materials science and the tissue engineering point of view.

## 2. Surface Functionalization Methods

### 2.1. Chemical Methods

Chemical surface functionalization leads to improved hydrophilicity on the fibers’ surface through changing atoms or molecules as a consequence of chemical reactions. Contrary to physical treatment, this kind of method is permanent. Once modified, the surface should stay permanently changed, by forming stable covalent bonding between biomolecules and the polymer surface. This is the effect of exposing relevant functional groups on the surface of the nanofibers to the external chemical stimuli [26]. To attain covalent bonding with biomolecules, usually, two steps are needed. The first one provides exposing functional groups such as –OH, –COOH, and –NH2 needed for effective reactions. The next one is formation of covalent bonds between biomolecules and functional groups on the polymer surface [6].

An advantage of chemical functionalization is that it provides steady bioactive sites for further biomolecules immobilization, resulting in improved biological properties of the surface, on which the cells can attach and grow favorably.

On the other hand, there is a risk that in the preactivated state, an uncontrolled chemical functionalization might change the bulk properties of the polymeric scaffold [21]. Figure 1 presents mechanisms of chemical surface modifications, while Table 1 summarizes the main pros and cons of the chemical surface functionalization methods applied to electrospun nanofibrous scaffolds.

#### 2.1.1. Wet Chemical Functionalization

Wet chemical functionalization of electrospun nanofibers is a chemical treatment under acid, or basic conditions, or with the presence of diamine, resulting in the polymer chains breaking at the sites of specific groups, like ester bonds in the case of polyesters.

Compared to the physical methods, wet chemical functionalization provides hydrophilic functionalization not only on the outer layer of the nanofiber surface, but also affects the deeper layer of the surface. The depth of reaction in wet chemical functionalization depends on the process/materials parameters like time reaction, and type and concentration of active substrates. Therefore, the reaction conditions must be precisely controlled [21,27] in order to minimize the changes of polymer bulk properties, which can ultimately lead to serious mechanical weakening or even destruction of polymer nanofibers [21,28]. It should be mentioned that wet chemistry usually uses harsh chemicals, making the method not ecologically friendly [29].

Depending on the method, –COOH, –OH, and –NH_2_ groups appear onto the polymer surface as a result of wet functionalization. The most common methods of wet chemical functionalization, aminolysis and hydrolysis, are described in detail below.

##### Aminolysis

Aminolysis [30] is a surface functionalization method that incorporates free amine groups to the nanofibers surface, by the formation of covalent bonds between the amine from the other end of diamine used for functionalization and the specific polymer group. The most common diamines used for functionalization of various polyesters, both aromatic and aliphatic, are 1,6-hexanediamine, ethylenediamine, or N-aminoethyl-1,3-propanediamine [31]. After effective aminolysis reaction at carefully controlled conditions, e.g., diamines concentration, temperature, pH, and time, the hydroxyl, carboxyl, and amine groups remain as active groups on the modified surface, providing better wettability. 

One of the advantages of the method is that the total time of the reaction is rather short: aminolysis can last from 2 up to 60 min [32,33]. The surface after the treatment is additionally characterized by higher roughness, favorable from the perspective of biomolecules immobilization and cells adhesion [21]. All of these features occur to be beneficial for both scenarios: when aminolysis is the ultimate treatment of surface functionalization ready for direct use, or while it serves as an intermediate step followed by immobilization of proteins such as collagen, fibronectin, and laminin, growth factors or bioactive groups. Adsorption of proteins on the aminolyzed surface can be performed via simple rinsing of the nanofibers for a certain period of time in the proteins solution [34,35].Although the whole procedure of physical adsorption of biological molecules seems to be relatively simple, there are some results indicating either insufficient concentration of –OH, –COOH, and –NH_2_ groups on the modified surface or proteins conformation, which does not enhance cell adhesion. Zhao et al. reported that in the case of PCL, the aminolyzed surface showed only temporal improvement of cell adhesion for 4 h [34].

Aminolysis as the intermediate step seems to be more effective than as the ultimate step, and is more often used in protein immobilization [6,36,37]. The efficiency of protein immobilization can be increased by additional treatment with a particular coupling agent, for instant, glutaraldehyde (GA), 1-ethyl-3-(3-dimethylaminopropyl)-carbodiimide (EDAC), Bis-N-succinimidyl-(pentaethylene glycol) (Bis (NHS) PEG5), or dimethyl adipimidate (DMA) [31,36]. Such coupling agents provide better availability of the –NH_2_ and –COOH on the surface, for covalent binding with biomolecules such as gelatin, chitosan, or collagen [28]. Depending on the choice of coupling agent, the functional group activation is carried out under specific pH, temperature, and time [31]. XPS, FTIR studies [31], and biological tests [34] have proved that such covalent binding between the surface and biomolecules significantly improves the morphology of the cells and promotes their proliferation. It indicates the importance of surface chemistry in cellular behavior on biomaterial substrates [34]. Aminolysis as the intermediate step is also widely used for growth factors immobilization. Haddad et al. have immobilized epidermal growth factors (EGF) to PLLA nanofibers through the coupling agent of (Bis (NHS)PEG5) [31]. Combining these treatments showed improved interactions between PLLA fibers and neural stem-like cells (NSLCs) after aminolysis in proliferation assay 2, 6, and 10 days [31].

Aminolysis might also be conjugated with other hydrophilic functionalization methods. Zhang et al. aminolyzed aligned PLLA fibers and then coated them with graphene oxide (GO) in the presence of the nerve growth factors (NGFs). Due to the phenol hydroxyl and epoxide groups on the basal plane and carboxylic groups at the edges, this treatment supposed to increase the hydrophilicity and growth and differentiation of various neural cell lines [37]. The water contact angle of the aligned PLLA decreased from 122.8 ± 1.5° to 96.8 ± 1.5° after aminolysis, and to 47.5 ± 2.2° after GO incorporation. The ultimate surface contact angle should be beneficial for cell growth accordingly to Webb’s studies [15] MTT proliferation and differentiation tests on PC12 and Schwann cells proliferation assay confirmed that, in the presence of NGF and appropriate hydrophilicity, the aminolyzed aligned PLLA/GO fibers effectively promoted Schwann cell growth and induced differentiation of PC12 [37].

Despite many advantages, aminolysis has its limitations. Its efficiency depends on the crystallinity of functionalized polymers, particularly on the surface crystallinity, as shown by Jeznach et al. [6]. The method is challenging for polymers with high crystallinity and small ratios of ester/alkyl groups, as in the PCL case. Moreover, aminolysis is more effective for films than nanofibers, which is related to the surface crystallinity resulting from the difference in crystallization conditions [6]. Thus, highly crystalline nanofibers require harsher conditions and extended time of the aminolysis reaction. Another limitation of the method is the instability of the aminolyzed surface. Zhu et al. [38] reported that aminolyzed polymers might lose amine groups from the surface while being stored at 37 °C. To avoid this, it is necessary to keep materials at a temperature below the polymer Tg. Amine groups also disappear from the surface while kept in the phosphate buffer saline (PBS) due to a restructuring of the polymer membrane [38]. Briefly, at the surface, segments of the polymer chains rearrange to decrease surface energy resulting in amine groups’ distancing from the surface. It constitutes a severe limitation for using of aminolyzed polymers in the industry or tissue engineering. Exceptionally, such scaffolds might be used very soon after fabrication or stored for a short time at very low temperatures.

##### Hydrolysis

The method is divided into alkaline and acid hydrolysis, which generally cleaves ester bonds to –COOH and –OH groups on the polyester’s surface [21]. Depending on the conditions, acid or alkaline, there are different mechanisms of the hydrolysis reaction. Similarly to aminolysis, both acid and basic hydrolysis, require to be conducted under precisely controlled conditions such as temperature, pH. and time, and the reaction time varies from a few to dozens of minutes [39].

The acid hydrolysis has an electrophilic character, which means that high-speed protons easily penetrate into uncharged polymer chains and are restored by the reaction. Briefly, carbon radicals are formed from the polymeric backbone, whereupon unsteady –OOH groups bind with O_2_ from H_2_O, resulting in –OOH groups breakdown into various groups such as –OH, –COOH, –R–O–R’, etc. [40]. Since –COOR groups are hardly available on the modified polymers’ surface, strong acidic conditions, elevated temperature, and appropriate time are required [31]. In the acid hydrolysis, mineral acids such as H_2_SO_4_, HClO_4_, and HCl and organic acids such as acetic acid (AA) or lactic acid (LA) are typically used [39,41,42]. Boland et al. chose 11.7 M HCl over NaOH for hydrolysis of PGA scaffolds to avoid a fiber diameter decrease. The hydrolysis of the –COO– exposed –COOH and –OH groups on the modified fiber surface, resulting in improved adhesion between surface and cells and their proliferation in the WST-1 proliferation assay on cardiac fibroblasts (FBs) after 4 days. However, in vivo studies on rat muscles were inconclusive and the whole functionalization procedure needs to be refined in the future. In other studies, Lee et al. used a mixture of HClO_4_/KClO_3_ in a saturated aqueous solution to modify PGA, PGLA, and PLLA scaffold surfaces. Although human chondrocytes and mouse NIH/3T3 fibroblasts showed an improved adhesion and proliferation after 2 days, this time might not be enough for further tissue engineering therapies [40]. An interesting phenomenon has been observed by Spinella et al. It is reported that in the case of some polymers such as cellulose nanocrystals, the use of H_2_SO_4_ for hydrolysis leads to a decrease in their thermal stability, leading to an uncontrolled release of H_2_SO_4_ from polymers during heating [43]. These features might limit the use of the acid hydrolysis in further studies on the surface functionalization of electrospun fibers for biomedical applications.

Due to the aforementioned limitations of the acid hydrolysis, for tissue engineering applications, the alkaline hydrolysis is more often chosen [43,44,45]. The alkaline hydrolysis has a nucleophilic character due to the presence of alkali metal hydroxides such as NaOH or KOH. The OH^−^ reacts with C=O, C–O, and C–O–C functional groups in the polymeric surface area, where there is the lowest electron density, and removes short segments of the polymeric chains. As a result, the hydrophilic groups such as –COOH and –OH are formed on the fibers’ surface [44]. The higher concentration of the alkali metal hydroxide the deeper the modification, and additional oxidizing agents accelerate the reaction, which causes OH^−^ penetration deeper into the material. On the other hand, very fast and hence uncontrolled hydrolysis leads to a decrease in the nanofiber mat’ mass and thickness [44]. Similarly to aminolysis, hydrolysis might also serve as an intermediate step before protein immobilization. Sadeghi et al. obtained PLGA scaffolds incorporated with collagen, applying hydrolysis as a 1st step followed by the 1-ethyl-3-(3-dimethylaminopropyl) carbodiimide/N-hydroxysuccinimide (EDC/NHS) crosslinking. The 1st step was carried out with the use of NaOH, in which fibers were hydrolyzed for 5 min at room temperature. Then, the fibers’ surface was activated through immersing fibers in EDC/NHS at pH = 6 at 4 °C. It was found that the wettability of PLGA increased significantly after hydrolysis and collagen immobilization, while after aminolysis, water contact angle decreased from 132° to 98° and further to zero after collagen immobilization. Additionally, FTIR spectra confirmed the presence of immobilized collagen on PLGA fibrous scaffolds, proving the effectiveness of the functionalization method. Cell viability MTT tests on human dermal fibroblasts (HDFs) and the keratinocyte cell line (HaCaT) showed increased viability of the cells on the scaffolds with immobilized collagen after 14 days, proving that modified in this way scaffolds are appropriate for skin tissue engineering applications [45]. In the other publication, De Luca et al. report on hydrolytically modified PLLA nanofibers after using KOH and NaOH for 1 h at room temperature. FTIR spectra confirmed the effectiveness of this method by showing the –COOH and –OH groups on the fibers’ surface [46]. The presence of these groups also affected the hydrophilicity of the PCL fibers, improving their biocompatibility. The water contact angle decreased from 76.58 ± 1.25° to 54.13 ± 2.73° and 56.83 ± 2.88° after NaOH and KOH hydrolysis, respectively. MTS proliferation assay on Schwann cells demonstrated a substantial increase in cell attachment and proliferation after 1, 3, and 5 days on modified samples compared to untreated fibers [46].

#### 2.1.2. Covalent Grafting

Covalent attachment or so-called “grafting” is generally carried out via chemical activation of appropriate reagents on the treated surface. Two major methods, “grafting to” and “grafting from”, can be distinguished in this regard.

“Grafting to” uses a coupling reaction that modifies the end functional polymer group with the reactive functional groups resulting in a chemical change in the polymeric backbone. This type of grafting uses all kinds of polymerization, including atom-transfer radical-polymerization (ATRP), controlled free-radical polymerization (CFRP), or anionic polymerization [47,48]. Covalent grafting was used for hydrophilic surface functionalization on PCL, PLLA, or polysulfone (PSU) and many copolymers described below [49,50,51,52]. “Grafting to” is a method that might be used for a wide range of materials, for instance, various monomers, such as acrylonitrile [22], acrylate, acrylamide, and many others [47,48,53]. However, the polymerization time has to be precisely controlled; otherwise, fibers’ morphology might be destroyed, and bulk properties of such material might be changed [54]. “Grafting to” might be combined with other methods of surface functionalization. Such a combination might bring not only hydrophilic surface functionalization but also provide additional features to the obtained fibers. Fu et al. combined technology of RAFT polymerization, ATRP, electrospinning, and “click chemistry” to obtain electrospun nanofibers made of poly (4-vinylbenzyl chloride) (PVBC), poly (glycidyl methacrylate) (PGMA), and poly-N-isopropylacrylamide (PNIPAM) to obtain fibers, which were solvent-resistant with the thermal-sensitive surface [48]. Contact angle measurements showed a significant change in PVBC74-b-PGMA46-g-PNIPAM7 fibers wettability depending on the ambient temperature. It was observed that the contact angle of hybrid fibers decreased from 140 to less than 30° after a temperature decrease from 45 to 20 °C [48].

In the field of hydrophilic surface functionalization even more popular than “grafting to” is “grafting from”. In this method, the macromolecular backbone is modified with the purpose of introducing reactive functional groups on the surface. This method requires an initiator, which might be introduced to the surface through post-polymerization, copolymerization, or polycondensation [51,52]. For instance, “grafting from” might take place using an atmospheric pressure plasma jet (APPJ) with argon as processing gas and in open-air. Maffei et al. functionalized PCL fibers with human vitronectin adhesive cue (HVP peptide). In this respect, NH_2_ groups were covalently grafted on the electrospun PCL mat by APPJ deposition of a coating using (3-aminopropyl) triethoxysilane (APTES) as a precursor. The fibers’ morphology was found unchanged. The MTT viability assay confirmed that after functionalization adhesion and migration of 533 osteoblasts increased, however, only after 2 h of culture. Further cellular tests are expected [50].

The other methods assume copolymerization grafting, induced biomolecules grafting, grafting polymerization of acrylic acid (AAc) on polyesters surface followed by a chemical reaction with biomolecules, or thermal-induced graft polymerization. Such functionalization usually takes place on surfaces that were previously treated with plasma [55,56,57]. Ma et al. [55] used copolymerization grafting of methacrylic acid (MAA) initiated by Ce (IV) after an air plasma treatment of electrospun PSU fibers. The plasma treatment of exposed oxygen-containing groups on the polymer surfaces, among which –OH groups combined with Ce^4+^ formed a redox initiating system, which initiated the graft polymerization of MAA. The modified surface was activated with toluidine blue O (TBO) and stable –COO– groups were formed and served as binding sites between proteins and functionalized surface. After that, bovine serum albumin (BSA) was incorporated into the surface. The BSA served as a substitute for the potential protein for further studies. BSA covalently immobilized on the PMAA grafted PSU fibers using the –COOH groups as coupling sites occurred to be successful. Obtained material served as a microfiltration membrane [55].

In the biomolecules grafting method, collagen type I (Col I), nanohydroxyapatite (nHA), fibrin, and chitosan are components that might be covalently incorporated into the polymer surface [56,57,58,59,60,61]. Chen et al. [62] treated PLLA nanofibers with oxygen plasma to expose –COOH groups on the surface and subsequently, by covalent grafting, cationized gelatin was incorporated into the fiber’s surface using an appropriate coupling agent (carbodiimide). After combining these two methods, the hydrophilicity of the surface significantly improved with the contact angle decreasing from 135 to 15° after PLLA functionalization. Gelatin grafting to the PLLA fibers also slightly changed the fiber morphology; they became rougher because of gelatin adhesion to the fibers. This change in the scaffold morphology occurred to be beneficial for chondrocytes, which spread not only on the surface of the modified fibers but also attached to the inner areas of the material’ structure. The MTS cell viability assay also confirmed that the modified surface created a more preferred environment for ECM production and chondrocytes proliferation even after 28 days [62]. Hesari et al. [51] incorporated plasma treatment with covalent grafting of gelatin to modify a PU scaffold. The surface was activated with oxygen plasma, and then gelatin macromolecules were incorporated into the surface. Combining these two methods did not significantly change the fibers’ microstructure, but improved hydrophilicity, through an increase of –NH_2_ and –COOH groups on the surface. Additionally, incorporated biomacromolecules influenced L929 fibroblasts’ behavior in the MTT assay after 7 days. Functionalized surface increased cytocompatibility of the material, improved fibroblasts spreading, and their proliferation ratio. Generally, grafting of reagents, such as gelatin, changes the microstructure of the fiber mat and increases the average pore size, which might be good from the cells’ perspective. On the other hand, it significantly impairs the mechanical properties of the material. Another thing is that gelatin increases the hydrolytic degradation rate, which might be favorable unless the process might be controlled [51]. On the other hand, surface graft polymerization usually requires plasma initiation to generate free radicals. Hence, conditions of the reaction have to be thoroughly controlled, otherwise undesirable side effects such as bulk properties deterioration, pore-blocking or collapsing, and fibers’ degradation [28] may take place.

**Table 1 polymers-12-02636-t001:** Comparison of chemical methods of hydrophilic functionalization of electrospun nanofibers.

Chemical Method	Mechanism	Advantages	Disadvantages
Aminolysis	Splitting of polymer chains by reacting with-NH_2_ groups and resulting introduction of active -NH_2_ and -OH on the surface, which may further be explored in secondary reactions to incorporate other functional groups [6,31]	short time of functionalization [6] increased roughness of the surface [6] non-toxicity of -NH_2_ groups in direct contact with cells, resulting in increased cells adhesion [33,34]	deep functionalization of the surface requires precise control of the conditions of the reaction [31] possible molecular degradation of polymer chains leading to mechanical weakness high crystallinity of the polyester limits effective modification [6] instability of amine groups on the surface at temperatures above Tg, especially when Tg is below physiological conditions [6] use of harsh chemicals making this method not ecofriendly
Hydrolysis	Cleavage of chemical bonds in polymeric chains by water molecules resulting in OH and COOH formation on the modified surface	a short time of functionalization increased roughness of the surface	deep and permanent functionalization of the surface requires precise control of the conditions possible molecular degradation of polymer chain leading to mechanical weakness use of harsh chemicals making this method not ecofriendly
Covalent grafting	Chemical functionalization of the polymer backbone to introduce reactive functional groups on the surface [47,48]	increase of average pore size resulting in better cell infiltration into the scaffold [51]	increase of average pore size resulting in a decrease in mechanical properties [51] requires precise control of functionalization time [28] uncontrolled hydrolytic degradation after biomolecules grafting [51] surface needs to be activated with plasma [51] biomolecules grafting might change the fibers microstructure

### 2.2. Physically/Chemically Functionalized Fibers

#### Plasma Treatment

Plasma treatment is a particular type of hydrophilic surface functionalization. It is usually categorized in the literature as a physical hydrophilic functionalization method [63,64]. However, plasma treatment results in significant chemical changes on the polymers surface such as chemical bonds breaking, leading to the introduction of various chemical groups like –OH, –COOH, CHO–, NH2–, –COO–, and reactive radicals, such as –COO (Figure 2) [65]. According to this, it could be classified as chemical functionalization [66]. Plasma treatment affects the surface energy of polymers and improves the wettability of the surface by changing their polarity. Type of the plasma source, time, and pressure are the main parameters controlling the functionalization process [23,26]. The method allows one to modify PGA, PLLA, PLGA, PCL, PEO, PVDF, PU, or polyaniline (PANI) electrospun mats, by forming appropriate functional groups such as –COOH on the modified surface as an effect of plasma glow discharge with O_2_ and C_3_H_4_O_2_ in the gaseous form [26,67,68,69]. Plasma (ionized gas) generates free radicals on the surface, which can behave similarly to polar groups [23]. Therefore, the following types of sources can be distinguished: argon, oxygen, methane [64,65,66,67,68,69,70], ammonia/helium, nitrogen, or air [19]. The plasma source might significantly influence the surface wettability, introducing different functional groups, which affect the immobilization of bioactive molecules on the treated surface. Asadian et al. [19] modified PCL fibers using various plasma sources such as oxygen, argon, ammonia/helium, or nitrogen showing a significant decrease in the contact angle from 135 before functionalization to 35° after argon plasma, to 24° after nitrogen plasma, and to 13° after He/NH_2_ plasma. The MTT assay on human foreskin fibroblasts (HFFs) revealed the significance of the discharged gas type on the cell viability. After 1 and 7 days of the cell culture, the number of viable cells was the highest for the argon plasma-treated samples and the lowest for the He/NH_3_ plasma-treated samples. It was explained as a result of the presence of O_2_-containing groups, which influenced more the cellular interaction than nitrogen functional groups after He/NH_3_ plasma functionalization.

Plasma treatment carries many advantages: it is ecofriendly and does not change bulk properties and hence mechanical properties of the fibers [71,72,73]. Compared to all chemical and physical hydrophilic functionalization methods, plasma treatment is the fastest method of surface functionalization and usually lasts from few seconds to few minutes [71]. Apart from all these advantages, plasma treatment effectively increases the hydrophilicity of hydrophobic polyesters modifying only the top of the fibers’ surface without affecting the fiber layers beneath. In the case of PLLA nanofibers modification using O_2_ plasma, Kooshki et al. [74] showed effective surface functionalization with –COOH and –OH groups resulting in a reduction of the contact angle from 135° to nearly zero [74]. Additionally, in vitro tests showed more significant mesenchymal stem cells (MSCs) attachment to the polymer surface and their enhanced proliferation in the MTT assay after 7 days.

Despite many advantages, plasma treatment has some disadvantages. Plasma does not affect the whole surface of the fibers, leaving some unmodified areas [28,31]. The method requires an appropriate plasma source and precise control of the modification time. Otherwise, the treatment might be ineffective or the morphology of fiber mats might be changed, even completely destroyed [75]. In some publications, it has been reported that plasma treatment might also worsen the mechanical properties of electrospun nanofibers [76]. Dolci et al., using atmospheric plasma for the PLLA nanofibrous scaffolds modification, observed the water contact angle reduction from 120 to 20°, and a drop in Young modulus from 86 ± 13 to 64 ± 8 MPa. Another limitation of this method is impermanence of the hydrophilic effect after functionalization, which fades away with time [77]. The reason is probably thermodynamic: a metastable system at a high free energy state proceeding towards a thermodynamically more stable state with lower free energy, which manifests in surface restructuring or so-called “surface aging” [78]. In effect, functional groups (polar groups) rearrange on the modified surface after being stored at room temperature. It was reported that storage conditions, type of polymer, and plasma treatment parameters significantly influence the stability of polar groups on the modified surface [79,80,81,82]. 

In order to prevent surface restructuring and extend the effect of plasma treatment, Wavhal et al. [83] recommend post-plasma grafting of acrylic acid (AA). This treatment allows for keeping hydrophilic groups on the polyethersulfone (PES) surface in an unchanged form for 2 months.

Undoubtedly, plasma treatment is a great method for surface activation; thereby, it is usually combined with other more permanent surface functionalization methods. Ghorbani et al. [69] activated the surface of PU-PANI scaffolds for bone regeneration with the use of oxygen plasma. The following step was PVA and PVA/3-glycidoxypropyltrimethoxysilane (GPTMS) immobilization to the activated surface. PVA immobilization allowed one to increase biocompatibility and amount of hydrophilic groups on the surface, while GPTMS provided better precipitation of hydroxyapatite and hence better osteoblasts adhesion to the modified surface. Atomic force microscopy (AFM) showed that after plasma treatment the degree of surface roughness increased from c.a. 97 to 144–429 nm. The waterdrop contact angle measurements showed a decrease from 116 for pure PU/PANI to 65° after plasma functionalization and to 62° after additional PVA/GPTMS immobilization, respectively. In vitro studies, using an MTT assay on MG-63 osteoblastoma cells, confirmed increased adhesion and proliferation of the cells on the functionalized surface after 7 days.

More examples of combining plasma treatment with other functionalization methods were described in the paragraph of covalent grafting and are discussed below.

### 2.3. Physically Functionalized Fibers by Physical Adsorption

Physical hydrophilic functionalization is an alternative to chemical methods and plasma surface functionalization (Figure 3). It allows overcoming some limitations of chemical methods. They do not change the bulk properties and therefore, mechanical properties, the process does not have to be precisely controlled: there are no risks of hydrolytic degradation, it does not require toxic solvents, which makes the method ecofriendly [84].

On the other hand, hydrophilic functionalization is based on forming the whole spectrum of physical interactions such as hydrophobic or electrostatic interactions, which usually are unstable and impermanent [24]. Moreover, physical methods might adversely affect the fibers morphology [85].

The most popular physical methods of hydrophilic surface functionalization are discussed in the following sections. Table 2 summarizes the main advantages and drawbacks of the physical surface functionalization methods on electrospun nanofibers.

#### 2.3.1. Simple Physical Adsorption

Physical adsorption is an uncomplicated method of hydrophilic and biological functionalization of electrospun nanofibers. The method assumes spraying or soaking of nanofibers in a biomolecules solution [24]. As a consequence of electrostatic, hydrogen, hydrophobic, and van der Waals interactions, biomolecules can adhere to the surface of the fibers [26,86]. Physical adsorption works for most nanofibrous polymers used in electrospinning such as PLGA, PEO, PLLA, poly (vinyl pyrrolidone)-iodine (PVPI), PCL, and PVA [87,88,89,90]. The most popular biomolecules adsorbed on the surface of the fibers are usually proteins, enzymes, polysaccharides, antibacterial agents, growth factors, or vitamins [24,26,91,92,93].

This kind of functionalization improves cell capability to recognize the surface receptors and increases their attachment to the modified surface without changing the bulk properties of the material scaffold [85]. Compared to chemical methods, or blending, physical adsorption protects biomolecules from denaturation in the presence of a harmful environment of organic solvents, high temperature, or high voltage during the electrospinning process [94]. Moreover, physical adsorption seems to be a simple method providing fast immobilization of the biomolecules to the nanofibers’ surface [89]. Nevertheless, some viscous bioactive agents such as hydrogels could change the fibers morphology significantly. Viscous solutions might fill most of the pores in the electrospun mats, blocking its availability for the cells. Esfahani et al. [85] modified polyamide-6 (PA6)/hydroxyapatite (HA) electrospun nanofibers with vitamin VD3 to enhance bone regeneration. Immersion in VD3 solution resulted in a dramatic increase in the fibers’ thickness, changes in fibers’ morphology, and blocking of pores after 1 h of immersion.

Another limitation of this kind of functionalization is impermanence since physical binding might be easily washed off the surface by polar solvents or the cell culture medium [24,95]. To extend the effectiveness of this method, resulting in enhanced cell affinity to the surface, it is necessary to combine two methods of surface functionalization. Yang et al. [24] conjugated plasma treatment and physical adsorption. PLLA samples were divided into modified with plasma and unmodified. Unmodified samples were directly soaked in collagen solution for 2 h, while modified samples were pretreated with ammonia plasma, followed by coating with collagen. The best effect has been seen while collagen adsorption was combined with plasma treatment. It increased the hydrophilicity of the fiber’s surface through the increased amount of –OH, –COOH, and –NH_2_ groups on the surface. It also facilitated collagen adsorption by forming of a strong polar interaction and hydrogen bonds between collagen and polar groups on the pretreated surface [24]. Conjugation of both functionalization methods increased affinity of 3T3 fibroblasts to the PDLA surface after 4 days, compared to the plasma-untreated fibers. In another paper, Jankowska et al. [96] modified poly (methyl methacrylate)/polyaniline PMMA/PANI with laccase enzyme by combining both adsorption and covalent binding methods to obtain potential membranes for separation processes or artificial scaffolds for medical applications. The first step of functionalization was the coupling reaction between 1-ethyl-3-[3-dimethylaminopropyl] carbodiimide hydrochloride/N-hydroxysuccinimide (EDC/NHS) and the PMMA/PANI surface. The second step was the physical adsorption of laccase. X-ray surface composition microanalysis confirmed that after combining covalent binding physical adsorption, the amount of oxygen, sulfur, chlorine, fluorine, and nitrogen increased significantly in comparison to the sample before surface functionalization. It was visible, especially in the content of sulfur and nitrogen, which increased almost twice after laccase binding.

#### 2.3.2. LBL Assembly

The LBL technique is a specific type of physical adsorption, which works on the principle of adsorption of oppositely charged polymers. By alternate deposition of opposite charge layers, it is possible to create self-assembled coating and obtain the desired properties of the composite material [26]. LBL was used for hydrophilic functionalization of PLLA, polyacrylonitrile (PAN), PLGA, PNIPAM, PCL, or PCL/silk fibroins [97,98,99,100,101,102,103]. LBL usually allows creation of layers made of polyelectrolytes, mainly natural materials such as proteins or polysaccharides. The layers might be positively or negatively charged depending on the conditions in the solution, for instant, by choice of pH [101]. In studies conducted by Li et al. [25], heparin (Hep) and chitosan (Cs) were adsorbed on the PCL/silk fibroin (SF) nanofibers as scaffolds for treatment of cardiovascular diseases. The first layer was made of positively charged Cs. The second layer was made of Hep, in which charge was changed to negative by adjusting pH of the solution. Strong layers adhesion was a consequence of electrostatic binding between Cs and Hep. Cs provided antibacterial properties, while Hep improved protein activity and stimulated the release of anticoagulant and fibrinolytic substances from vascular endothelial cells. The Alamar blue test confirmed good biocompatibility and improved cell proliferation of modified PCL/SF scaffold. It was also demonstrated, that after LBL, human umbilical vein endothelial cells (HUVEC) showed improved adhesion and proliferation on the modified surface after 48 h. The antibacterial test showed that adsorbed layers reduced infectivity with *Escherichia coli* and *Staphylococcus aureus* up to 95% [25]. In another study, Zhang et al. [102] deposited two layers: the self-assembling peptide (SAP) and RGD sequences on PCL nanofibers for neural tissue engineering applications. Adhesion between the first SAP layer and PCL surface was a consequence of strong hydrophobic interactions. Plasmid DNA (pDNA) complexes were used as a delivery system of the CRISPR/dCas9 gene expression control system and with positively charged SAP and negatively charged RGD was adsorbed as a consequence of electrostatic interactions. Deposited layers provided sustained release of pDNA, enhanced cell adhesion, and proliferation of human mesenchymal stem cells (hMSCs) and human neural progenitor cells (hNPCs) in Quant-iTPicoGreen dsDNA assay and Click-iTEdU cell proliferation assay after 7 days of culturing. Using this method for nanofibers functionalization also occurred to be a useful tool for non-viral genome editing. 

LBL shows many advantages, one of them is that it does not change the bulk properties of nanofibers [103]. From the processing point of view, LBL is easy to conduct, universal for covering various even complex structures, and a wide range of charged substrates might be used in this technique [26]. Similarly to the physical adsorption, deposited biomolecules are protected from denaturation, avoiding the loss of provided functions [104].

The main limitation of this method is that both modified fibers surface and deposited layers always need to be charged. For this reason, before LBL functionalization, the surface has to be prepared by providing an appropriate charge. Chemical methods such as aminolysis, sulfonation of phenyl groups (if they are present in the polyester material), and alkaline hydrolysis are examples of methods providing an appropriate charge on the surface before LBL treatment [97,98,105]. Wang et al. [106] functionalized the surface of PCL aligned fibers through grafting amino groups with the use of polyethyleneimine (PEI) for 36 h. The next step was the LBL deposition of nanocoatings made of antibacterial drug poly-L-lysine (PLL) encapsulated in gelatin and heparin. According to FTIR analysis, deposited layers showed effective adhesion to the polymers surface as a consequence of electrostatic interactions, where the gelatin –COOH groups bound strongly with the PLL –NH_3_ groups. Additionally, there were electrostatic interactions between the heparin –COOH groups and the PLL –NH_3_ groups, and between heparin and gelatin. LBL coating, especially gelatin, was crosslinked with genipin (GNP), and at the end, MMP2 enzyme was adsorbed on the layers in order to induce the controlled release of PLL. SEM images showed that after GNP crosslinking, the fibers lost their morphology, and GNP presence most likely blocked the action of the enzyme through an uncoupling protein and reaction with free amino groups. Cell proliferation CCK-8 assay on HUVEC showed significant improvement in the cells amount on the scaffold after functionalization after 3 and 5 days. However, it was demonstrated that the layer of Gelatin/PLL had the greatest influence on the increased cell affinity to the surface. It was an effect of the presence of bioactive polypeptides of gelatin and positive charge of PLL, which promoted HUVECs increased attachment and proliferation [106].

**Table 2 polymers-12-02636-t002:** Comparison of electrospun nanofibers physical hydrophilic functionalization methods [24,26,85,93,97,98,104,106].

Physical Method	Mechanism	Advantages	Disadvantages
Simple physical adsorption	Weak physical interactions such as hydrophobic interactions, hydrogen bonds, van der Waals interactions [24,26]	does not change bulk properties of the polymer [93] protects biomolecules from challenging environment simple, universal	might change fibers morphology, for instance increases fibers thickness or clogs the pores [85] impermanent [24]
LBL	Electrostatic interactions as an effect of alternate embedding of oppositely charged substances [26]	does not change the bulk properties of polymer protects biomolecules from a challenging environment [104] simple, universal [26]	only charged substances might be used [98,106] modified surface needs to be charged, or previously pre-treated to deposit charge on the surface [97]

## 3. Tissue Engineering Applications of Functionalized Polymer Nanofibers

Electrospun nanofibrous scaffolds possess many interesting properties for tissue engineering applications, such as high surface area, and morphology, which fairly mimic ECM. Native ECM is responsible for the stimulation of cellular mobility and further migration. An appropriate interaction between fibrous ECM and the cellular cytoskeleton leads to improved cell mobility. Cells’ ability to recreate their own ECM is crucial for effective tissue regeneration [107]. However, besides appropriate morphology, the perfect scaffold should also have appropriate biochemical cues, which might be obtained by surface chemistry modification. Electrospun nanofibers made of aliphatic polyesters show relatively poor hydrophilicity, and consequently, low biocompatibility [108]. Such materials without surface functionalization do not have functional groups, which might be recognized by cell-binding receptors increasing cell-scaffold adhesion.

An increase in hydrophilicity of electrospun membranes is the first step before immobilization of biomacromolecules such as proteins or growth factors, providing stable anchor points for the cells. The cells in contact with the scaffold need to feel materials surface biochemistry, which might promote cell-signaling pathways. One of the positive effects of biologically functionalized material is guiding the transcription factors responsible for the cell’s fate and regulation of their differentiation [13]. Surface modification with a protein, such as gelatin, which is extensively involved in building of ECM, not only increases hydrophilicity but also mostly gives a cellular response. It is a result of the recognition of bioactive ligands of RGDsequences by integrins.

Another protein containing RGD sequences is silk fibroin [109]. Bhattacharjee et al. aminolyzed PCL nanofibers and activated the functional groups on the surface with GA, with subsequent silk fibroin immobilization. MTT viability assay and Alamar blue proliferation assay on osteoblast-like MG 63 cells showed not only increased cell growth and adhesion to the functionalized surface (Figure 4), but the cells were also able to create the native ECM, leading to bone tissue regeneration. Similarly to these studies, a different bioactive protein—collagen was immobilized to PCL fibers previously functionalized by aminolysis [110]. Biological tests on NIH 3T3 fibroblasts showed not only increased proliferation on functionalized scaffolds but also enhanced infiltration inside the scaffolds. An increase in cell proliferation was explained as a result of collagen presence, while enhanced infiltration most likely as a result of increased hydrophilicity, which influenced significantly cell–scaffold interactions.

The same method has been also proven in the case of scaffolds for neural regeneration. Amores et al. [111] used laminin and RGD-containing peptide GRGDSP as the immobilized biomolecules to the PCL scaffold. Proliferation tests using neural stem cells showed increased cells’ density on the functionalized scaffolds due to good access of the cell integrin receptor to the laminin and RGD sequences on the scaffolds surface. Most importantly, neural stem cells were able to differentiate into neurons and astrocytes showing enhanced neurons alignment on the functionalized material. These examples have proven that proper scaffold morphology or mechanical properties alone are not sufficient for effective tissue regeneration. More challenging tissues, such as neural, seem to need a more sophisticated approach, such as hydrophilic functionalization followed by biological groups incorporation.

## 4. Conclusions and Future Perspectives

The review discusses current methods used for hydrophilic surface functionalization of electrospun nanofibers targeting medical applications. Surfaces of nanofibrous mats can be functionalized with wet chemical methods, covalent grafting, plasma treatment, and physical methods based on physical adsorption. It may be concluded that a single hydrophilic functionalization method results in serious limitations, thus, an alternative approach is required. It may be expected that future strategies would be a combination of two or even more hydrophilic functionalization methods, at least one being chemical, for instance, covalent grafting, aminolysis, or oxygen plasma followed by physical LBL assembly. For instance, a combination of aminolysis, or hydrolysis with biomolecules immobilization, not only improves hydrophilicity of the surface but also significantly increases aliphatic polyester scaffolds biocompatibility. Both hydrophilicity and biomolecules incorporation are emerging important issues in tissue engineering applications. The most effective functionalization methods, ultimately, should provide sufficient surface biological activity, integrity, and mechanical properties of the scaffold. Future strategies might also include additional specific benefits for defined applications, for instance, electrical conductivity in scaffolds devoted to neural tissue engineering.

Future strategies combining more than one functionalization method are foreseen as highly important for further development of electrospun nanofibers for tissue engineering applications. We believe that combining and developing of hydrophilic functionalization methods will widely use functionalized electrospun nanofibers in clinical applications, and this is only a question of time.

## Figures and Tables

**Figure 1 polymers-12-02636-f001:**
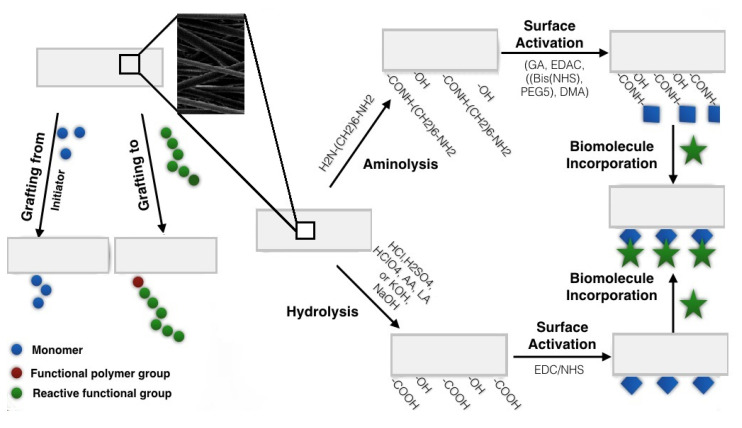
Chemical functionalization of electrospun nanofibers.

**Figure 2 polymers-12-02636-f002:**
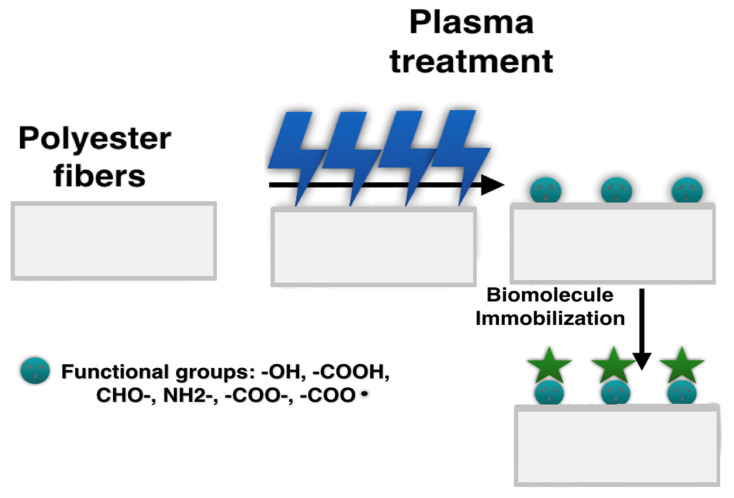
Plasma treatment of electrospun nanofibers.

**Figure 3 polymers-12-02636-f003:**
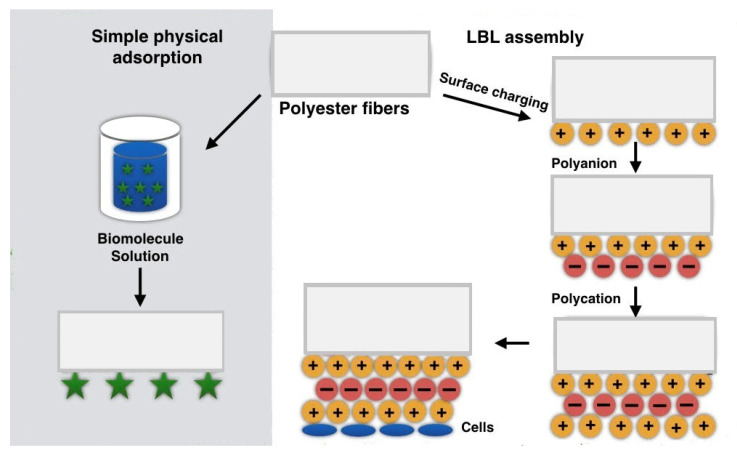
Physical functionalization of electrospun nanofibers.

**Figure 4 polymers-12-02636-f004:**
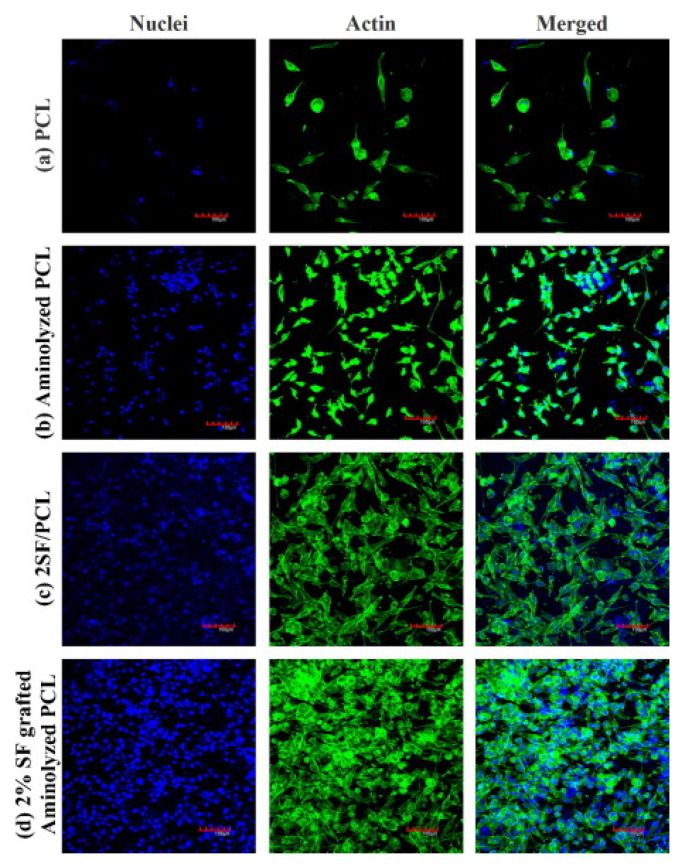
The cytoskeletal actin organization and distribution of MG-63 cells grown on (**a**) unmodified PCL nanofibers, (**b**) PCL after aminolysis, (**c**) PCL blended with silk fibroin (SF), and (**d**) aminolyzed PCL after subsequent SF immobilization at day-point 7. Reprinted from *Eur. Polym. J.* 2015, 71, 490–509 with permission from Elsevier [109].

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
