# Peer review of "Hydrophilic Surface Functionalization of Electrospun Nanofibrous Scaffolds in Tissue Engineering"

_polymers, 2020, doi:10.3390/polym12112636_

Round 1

Reviewer 1 Report

It is quite good review paper in one of possible field of electrospun webs application and presents problem very well known for researchers. It could be good review paper and source of information for younger researchers which starts their activity in present field.

Author Response

Dear Reviewer,

Thank you for your valuable comments. We did our best to improve the manuscript’s quality to present a reliable source of information for researchers who start their activity in the present field.

Best regards,

Beata Niemczyk-Soczynska, Arkadiusz Gradys, and Pawel Sajkiewicz

Reviewer 2 Report

  1. The author list many common techniques in “2. Surface functionalization methods”. I think they are not important to “electrospun nanofibrous” in this mini-review, but occupy too much space.
  2. All the figures use comb structure to present the materials surface. Personally, I think this structure is too strange.
  3. There is no deep discussion and comparison on how the “electrospun nanofibrous scaffolds” advanced hydrophilic surface and how this technique be developed and benefit to “tissue engineering”.

Author Response

Dear Reviewer,

Thank you for your valuable comments and suggestions. According to your comments, we have made corrections to the manuscript. Our answers and changes done in the text are listed below.

  1. The author list many common techniques in “2. Surface functionalization methods”. I think they are not important to “electrospun nanofibrous” in this mini-review, but occupy too much space.

Answer: Thank you for your valuable suggestion. However, in this respect, the authors would like to keep paragraph 2 in an unchanged version. Paragraph 2 contains important information from the perspective of materials science, creating this publication’s entire point. The authors would like to extensively discuss various hydrophilic surface functionalization methods, considering the advantages and disadvantages of the described methods. Therefore, this review should be rather considered as an in-depth review than a mini-review. We aim to create a reliable source of information for researchers who start their activity in both or one of the current fields: materials science as well as tissue engineering.

  1. All the figures use comb structure to present the materials surface. Personally, I think this structure is too strange.

Answer: The comb structures that are supposed to mimic the surface of the materials have been changed into more adequate in all of the figures.

  1. There is no deep discussion and comparison on how the “electrospun nanofibrous scaffolds” advanced hydrophilic surface and how this technique be developed and benefit to “tissue engineering”.

Answer: To discuss how hydrophilic surface modification improves electrospun scaffolds’ properties in tissue engineering, an additional section entitled:” Tissue engineering applications of functionalized polymer nanofibers” has been added on page 16.

Reviewer 3 Report

In this manuscript, the authors summarized recent advance in the hydrophilic surface functionalization of electrospun polymer nanofibers, in which the chemical, physical/chemical, and chemical methods for functionalization were introduced and discussed in detail. In addition, some effects of the surface functionalization on tissue engineering application of nanofibrous scaffolds were demonstrated. It is an interesting and important work to the fields of polymer science, surface science, biomedicine, and others. I recommend its publicatio at Polymers after minor revisions.

Special comments:

  1. It is suggested for the authors to provide more information on the novelty and significance of this manuscript, for instance, by comparing with other previously review papers on electrospinning polymer nanofibers.
  2. There are only a few figures in this review, which are not enough for readers to get full understanding on the functionalization of polymer nanofibers. Therefore, it will be better if the authors could provide a few more cases with nice pictures on some important references.
  3. It is suggested for the authors to add a section on the "tissue engineering applications of functionalized polymer nanofibers", which will make this review more interesting and attractive by readers in biomaterial field.
  4. More discussion on the future perspectives of this research topic should be given.

Author Response

Dear Reviewer,

Thank you for your valuable comments and suggestions. We did our best to improve the manuscript’s quality. According to your comments, we have made corrections to the manuscript. Our answers and changes done in the text are listed below.

  1. It is suggested for the authors to provide more information on the novelty and significance of this manuscript, for instance, by comparing with other previously review papers on electrospinning polymer nanofibers.

Answer: Some corrections have been done on page 2, line 73. This manuscript's significance is extensive discussion on various hydrophilic surface functionalization methods, considering the advantages and disadvantages of the described methods. The aim is to create a reliable source of information for researchers who start their activity in both or one of the current fields: materials science and tissue engineering. According to the authors' knowledge, this field hasn’t been discussed in such an in-depth manner as in our review article. 

  1. There are only a few figures in this review, which are not enough for readers to get full understanding on the functionalization of polymer nanofibers. Therefore, it will be better if the authors could provide a few more cases with nice pictures on some important references.

Answer: The authors provided Figure 4. from one of the important references.

  1. It is suggested for the authors to add a section on the "tissue engineering applications of functionalized polymer nanofibers", which will make this review more interesting and attractive by readers in biomaterial field.

Answer: To improve the article’s quality an additional section entitled:” Tissue engineering applications of functionalized polymer nanofibers” has been added on page 16.

  1. More discussion on the future perspectives of this research topic should be given.

Answer: Discussion on future perspectives has been given in sections 3 and 4.

Round 2

Reviewer 2 Report

Thank you so much for the author directly reply to my question. Here are three comments for author to develop this review:

  1. All these techniques describe in the manuscript didn’t point out the importance to the topic as author mentioned in title.

I didn’t see any valuable information or discussion in this part to support how these common techniques are significant to influence the “electrospun nanofibrous scaffolds” surface functionalization as with bulk materials.

These techniques have been vast review and summarized in many material-based surface functionalization comprehensive reviews with deep discussion for biomedical applications. It is not so necessary to repeat the old topic without new perspective view and published as mini-review on Polymer

  1. The hydrophilic surface is not the final goal to solve tissue engineering problem.

In “2. Surfae functionalization method”, all the chemical or physical method is not simply for the aim to increase the hydrophilic property. It is not simply to induce -COOH, -OH or -NH2 as surface conjugation site to increase the surface free energy and improve the hydrophilic. The most important thing is based on the biomolecules’ intrinsic property to solve the tissue engineering problem.

  1. The hydrophilic surface is problematic to the tissue engineering in some conditions.

For example, many hydrophobic drugs, biomacromolecules with hydrophobic interaction have been modified to surface to solve the tissue engineering problem and the hydrophilic property significant decrease significantly, but I didn’t see any discussion.